

# Complete chloroplast genome assembly and phylogenetic analysis of blackcurrant (*Ribes nigrum*), red and white currant (*Ribes rubrum*), and gooseberry (*Ribes uva-crispa*) provide new insights into the phylogeny of Grossulariaceae

Xinyu Sun[1], Ying Zhan[1], Songlin Li[1], Yu Liu[1], Qiang Fu[1], Xin Quan[1], Jinyu Xiong[1], Huixin Gang[1,2,3], Lijun Zhang[2,4], Huijuan Qi[2,4], Aoxue Wang[1,3], Junwei Huo[1,2,3], Dong Qin[1,2,3] and Chenqiao Zhu[1,2,3]

[1] College of Horticulture & Landscape Architecture, Northeast Agricultural University, Harbin, Heilongjiang, China
[2] National-Local Joint Engineering Research Center for Development and Utilization of Small Fruits in Cold Regions, National Development and Reform Commission, Harbin, Heilongjiang, China
[3] Key Laboratory of Biology and Genetic Improvement of Horticultural Crops, Ministry of Agriculture and Rural Affairs, Harbin, Heilongjiang, China
[4] Heilongjiang Institute of Green Food Science, Harbin, Heilongjiang, China

Corresponding authors
Dong Qin, dongq9876@126.com
Chenqiao Zhu, zhu8937zhu@126.com

## ABSTRACT

**Background:** Blackcurrant (*Ribes nigrum*), red currant (*R. rubrum*), white currant (*R. rubrum*), and gooseberry (*R. uva-crispa*) belong to Grossulariaceae and are popular small-berry crops worldwide. The lack of genomic data has severely limited their systematic classification and molecular breeding.
**Methods:** The complete chloroplast (cp) genomes of these four taxa were assembled for the first time using MGI-DNBSEQ reads, and their genome structures, repeat elements and protein-coding genes were annotated. By genomic comparison of the present four and previous released five *Ribes* cp genomes, the genomic variations were identified. By phylogenetic analysis based on maximum-likelihood and Bayesian methods, the phylogeny of Grossulariaceae and the infrageneric relationships of the *Ribes* were revealed.
**Results:** The four cp genomes have lengths ranging from 157,450 to 157,802 bp and 131 shared genes. A total of 3,322 SNPs and 485 Indels were identified from the nine released *Ribes* cp genomes. Red currant and white currant have 100% identical cp genomes partially supporting the hypothesis that white currant (*R. rubrum*) is a fruit color variant of red currant (*R. rubrum*). The most polymorphic genic and intergenic region is *ycf1* and *trnT-psbD*, respectively. The phylogenetic analysis demonstrated the monophyly of Grossulariaceae in Saxifragales and the paraphyletic relationship between Saxifragaceae and Grossulariaceae. Notably, the *Grossularia* subgenus is well nested within the *Ribes* subgenus and shows a paraphyletic relationship with the co-ancestor of *Calobotrya* and *Coreosma* sections, which challenges the dichotomous subclassification of the *Ribes* genus based on morphology (subgenus *Ribes* and

subgenus *Grossularia*). These data, results, and insights lay a foundation for the phylogenetic research and breeding of *Ribes* species.

# INTRODUCTION

*Ribes* is the sole genus in the Grossulariaceae family and comprises approximately 150 to 200 known species. *Ribes* species are deciduous perennial shrubs that are native to temperate areas of Eurasia, North America, the northwestern part of Africa, and the Andes (*Cortez & de Mejia, 2019*). Cultivated *Ribes* species, commonly referred to as "currant" and "gooseberry", are popular and old small berry crops in Europe and Asia (*Lukša et al., 2018*; *Orsavová et al., 2019*). Blackcurrant (*R. nigrum*), red currant (*R. rubrum*), white currant (*R. rubrum*), and gooseberry (*R. uva-crispa*) (Fig. 1) are dominant cultivated taxa due to their high yield, fruit quality, and ecological adaptability (*Hummer & Dale, 2010*). In 2021, the global cultivation area of currants and gooseberries was 148,275 hectares, with a fruit production of 0.82 million tons, reaching a gross value of 1,503 million USD (FAOSTAT, https://www.fao.org/faostat/en/#data). Currants and gooseberries are commonly processed into juice, wine, candy, and jam owing to their high concentrations of functional metabolites and distinctive flavors (*Määttä, Kamal-Eldin & Törrönen, 2001*). In recent years, they have been reported to contain multi-functional bioactive metabolites, such as anthocyanins, polyphenols, phenolic acids, and flavonoids, and are characterized by potential antioxidant, anti-inflammatory, and antibacterial bioactivities that are involved in antidiabetic, antineoplastic, anti-tumor, anti-cardiovascular, and anti-neurodegenerative functions (*Lukša et al., 2018*; *Orsavová et al., 2019*). Despite their great economic importance and health potential, there has been little research on the *Ribes* species to date, which not only lags far behind that on major fresh fruits such as apple, citrus, and grape, but also falls behind that on emerging fruit crops that are mainly supplied for the processing market, such as sea-buckthorn (*Wu et al., 2022*; *Yu et al., 2022*), blue honeysuckle berry (*Zhu et al., 2023*), and wolfberry (*Cao et al., 2021*). In particular, the phylogeny of Grossulariaceae and the infrageneric relationships of the *Ribes* genus remain highly controversial (*Weigend, 2007*).

Based on morphology, *Ribes* was once included in the polyphyletic family Saxifragaceae (*Lingdi & Alexander, 2001*) but is now more accepted as the sole genus in the Grossulariaceae family (*Cronquist & Takhtadzhian, 1981*). However, because some literature still places Grossulariaceae under Saxifragaceae, the delineation of the Grossulariaceae family remains blurry (*Messinger, Hummer & Liston, 1999*), and its monophyletic origin has been rarely discussed in previous studies because only a limited number of *Ribes* species were used (*Dong et al., 2013*). In addition, the infrageneric classification of *Ribes* remains unclear due to its wide geographical distribution, unknown interspecific hybridization, and high phenotypic diversity (*Weigend, 2007*; *Zhang et al., 2023*). From the first monograph of Janczewski on *Ribes* (*Janczewski, 1907*), the

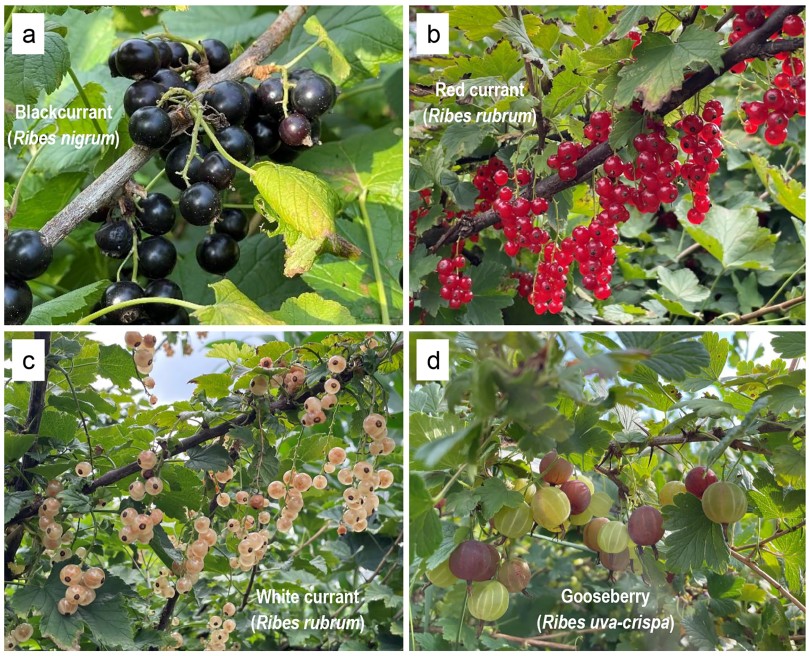

**Figure 1 Morphology of the four *Ribes* taxa.** (A) Blackcurrant (*R. nigrum*). (B) Red currant (*R. rubrum*). (C) White currant (*R. rubrum*). (D) Gooseberry (*R. uva-crispa*). Photo credit: by Xinyu Sun.                                                      

infrageneric classification of *Ribes* has been constantly proposed and revised (*Coville et al., 1908*; *Berger, 1924*; *Poyarkova, 1939*; *Rheder, 1940*; *Sinnott, 1985*; *Weigend, 2007*), resulting in a highly confusing taxonomy. In the work of Janczewski, the *Ribes* genus was divided into six subgenera with eleven sections (*Grossularioides*, *Parilla* (*Parilla* and *Andina*), *Berisia* (*Diacantha* and *Euberisia*), *Coreosma* (*Coreosma*, *Symphocalyx*, *Heritiera*, *Cerophyllum* and *Calobotrya*), *Ribes*, and *Grossularia* (*Grossularia* and *Robsonia*)) (*Janczewski, 1907*). Moreover, Berger divided Grossulariaceae into the *Ribes* genus (currants, including eight subgenera) and the *Grossularia* genus (gooseberries, including four subgenera). This classification was later demoted to two corresponding subgenera with 12 sections, which is the most consensus and widely accepted classification of *Ribes* so far (*Coville et al., 1908*; *Berger, 1924*; *Poyarkova, 1939*; *Sinnott, 1985*; *Messinger, Hummer & Liston, 1999*). Namely, the *Ribes* subgenus includes eight sections: *Berisia* (alpine currant), *Calobotrya* (red-flowering currant), *Coreosma* (blackcurrant), *Grossularioides* (spiny currant), *Heritiera* (skunk currant), *Parilla* (Andine currant), *Ribes* (red currant), *Symphocalyx* (golden currant), and the *Grossularia* subgenus includes four sections: *Grossularia* (true gooseberry), *Robsonia* (Sierra gooseberry), *Hesperia* (ornamental gooseberry), and *Lobbia* (desert gooseberry). Although internal transcribed spacer, nontranscribed spacer, and chloroplast markers have been recently applied to resolve the infrageneric phylogeny of *Ribes* (*Messinger, Hummer & Liston, 1999*; *Senters & Soltis, 2003*; *Schultheis & Donoghue, 2004*), inconsistent results have been reported regarding the monophyly of gooseberries (subgenus *Grossularia*) and the phylogenetic relationship among the above-mentioned 12 sections. Additionally, the genomic data of *Ribes*
have been exclusively used in studies of the evolution of Saxifragales (*Dong et al., 2013*; *Wang et al., 2021*) and the genetic diversity of endemic *Ribes* populations (*Zhang et al., 2023*).

Plastids are semi-autonomous organelles in the plant cell, carrying out photosynthesis, and various metabolic and signaling functions (*Daniell et al., 2016*; *Dobrogojski, Adamiec & Lucinski, 2020*). Due to the haploidy, uniparental inheritance, no genetic recombination, conserved genomic structure, and relatively small genome size of plastids, the polymorphic loci of plastids have been increasingly applied to plant phylogenetics, taxonomy, population genetics, and marker development for molecular breeding and DNA barcoding (*Olmstead & Palmer, 1994*; *Kuang et al., 2011*; *Dong et al., 2012*; *Gao et al., 2022*). Meanwhile, the possibility that a lack of variation in the modest number of plastid genes may hamper species discrimination, suggesting improved resolutions based on complete plastome sequences (*Parks, Cronn & Liston, 2009*; *Nock et al., 2011*). Through comparative analysis based on the complete chloroplast (cp) sequences, the phylogeny of some plants, such as *Cardiocrinum*, *Ficus*, Orchidaceae, and Myrtales, has been delineated from the species level to the order level (*Zhang et al., 2021*; *Chen, Hu & Zhang, 2022*; *Huang et al., 2022*; *Jiang et al., 2022*). However, only five *Ribes* cp genomes have been released in NCBI to date (Jan. 2023). Among them, *R. glaciale* and *R. fasciculatum* var. *chinense* (*Dong et al., 2018*) are wild species indigenous to East Asia; and Sierra currant (*R. nevadense*), Sierra gooseberry (*R. roezlii*) (*Folk et al., 2020*), and clove currant (*R. odoratum*) (*Wang et al., 2021*) are endemic species of America. There has been no report about the cp genomes of cultivated *Ribes* species and their phylogenies, which has severely limited the germplasm exploration, genetic conservation, cross-breeding, and genetic improvement of *Ribes*.

Herein, complete cp genomes of blackcurrant (*R. nigrum*), red currant and white currant (*R. rubrum*), and gooseberry (*R. uva-crispa*) were assembled and characterized, laying a molecular foundation for the research on *Ribes* taxa. The study also identified the highly polymorphic sites across all the nine released *Ribes* cp genomes, providing practical tools for molecular breeding and DNA barcoding. Moreover, phylogenetic trees of Saxifragales were constructed by using 31 complete cp genomes, which can shed new light on the phylogeny of the Grossulariaceae family and *Ribes* species.

## MATERIALS AND METHODS

### Plant materials and DNA extraction

The fresh leaves of blackcurrant 'Liangye houpi' (*R. nigrum*), red currant 'Red Cross' (*R. rubrum*), white currant 'Witte Parel' (*R. rubrum*), and gooseberry 'Pixwell' (*R. uva-crispa*) were collected at the Horticultural Station of Northeast Agricultural University (126.73°E, 45.74°N), Harbin, China. Samples were frozen immediately in liquid nitrogen and kept at an ultra-low temperature freezer (–80 °C) until DNA extraction. Total genomic DNA was isolated using the Hi-Fast Plant Genomic DNA Kit (GeneBetter BioTech Co. Ltd., Beijing, China) following the manufacturer's instructions (http://www.gene-better.cn/).

## Sequencing, assembly, and annotation

The DNA library preparation and next-generation sequencing were accomplished at *Frasergen Bioinformatics* Co., Ltd. (Wuhan, China) following the manufacturer's instructions (MGI Tech Co., Ltd., Shenzhen, China) of the DNBSEQ-T7 platform. The adaptors of the raw reads were removed using Trimmomatic v0.93 (*Bolger, Lohse & Usadel, 2014*) and then the reads were filtered using SOAPnuke v2.X (*Ashraf et al., 2018*) with parameters of "–lowQual=20, –nRate=0.005, –qualRate=0.5". The four cp genomes were assembled using GetOrganelle v1.7.6.1 (*Jin et al., 2020*) with parameters of "-R 15 -k 21,45,65,85,105" and with the reference of *R. fasciculatum* var. *chinense* (MH191388). Genomic annotation was performed using CPGAVS2 (*Shi et al., 2019*) and further examined manually according to *R. nevadense* (MN496075) and *R. roezlii* (MN496076) (*Folk et al., 2020*). The cp feature map was constructed using OGDRAW v1.3.1 (*Greiner, Lehwark & Bock, 2019*) with default parameters.

## Comparative genomic analysis

The five previously reported *Ribes* cp genomes were re-annotated using CPGAVS2 with the reference of *R. fasciculatum* var. *chinense* (MH191388). The nine *Ribes* cp genomes were aligned using MAFFT v7.471 (*Katoh & Standley, 2013*). The pairwise genetic divergence was calculated using BioEdit v7.2.6 (*Hall, Biosciences & Carlsbad, 2011*). The IR boundary comparison of the nine *Ribes* was performed using IRSCOPE (*Amiryousefi, Hyvönen & Poczai, 2018*). mVISTA program (*Frazer et al., 2004*) was used to visualize the alignments of the nine *Ribes* cp genomes with the reference of *R. fasciculatum* var. *chinense*.

## Repeat, SNP, and indel identification

Tandem Repeats Finder (*Benson, 1999*) was used to determine tandem repeats with default parameters. REPuter (*Kurtz et al., 2001*) was used to determine forward (F), palindromic (P), reverse (R), and complement (C) repeats with the parameters of "Hamming Distance: 3; Maximum Computed Repeats: 5,000 bp; Minimum Repeat Size: 30 bp". SSR loci were identified using Misa-web (*Beier et al., 2017*) (http://pgrc.ipk-gatersleben.de/misa/) with default parameters. SNPs and Indels among the nine cp genomes were identified using DnaSP v6.12 (*Rozas et al., 2017*) and the *Pi* values were calculated with a window length of 600 bp and a step size of 200 bp.

## Phylogenetic analysis

The complete cp genomes of 20 representative species in Saxifragales, four focal taxa in the present study, five previously released *Ribes* species, and two outgroup species (Rosales) were selected for phylogenetic analysis and their Genbank accession numbers are shown in Table S10. The multi-sequence alignment was formalized using trimAL (*Capella-Gutiérrez, Silla-Martínez & Gabaldón, 2009*). The maximum-likelihood (ML) phylogenetic tree was constructed using MEGA v11.0.13 in the multithread mode with 1,000 bootstrap (BS) replications (*Tamura, Stecher & Kumar, 2021*) and further annotated using iTOL v6 online (https://itol.embl.de/) (*Letunic & Bork, 2021*). The Bayesian tree was constructed

using BEAST 2.7.5 (*Bouckaert et al., 2019*) following the standard workflow (https://beast2. blogs.auckland.ac.nz/tutorials/) and referenced the literature reported by *Nascimento, dos Reis & Yang (2017)*. The confidences of the clades were evaluated by using posterior probabilities (PP).

## RESULTS

### Organizations and gene features of chloroplast genomes of four *Ribes*

The cp genomes of the four *Ribes* taxa ranged from 157,450 bp (*R. nigrum*) to 157,802 bp (*R. rubrum*) in length (Fig. 2). The large single copy (LSC), small single copy (SSC), and inverted repeat regions (IRa/IRb) of the four cp genomes ranged from 87,086 bp (*R. nigrum*) to 87,369 bp (*R. uva-crispa* and *R. rubrum*), 18,288 bp (*R. uva-crispa*) to 18,423 bp (*R. rubrum*), and 26,001 bp (*R. uva-crispa*) to 26,007 bp (*R. nigrum*), respectively (Table 1). The coding sequence (CDS) length of the four cp genomes was from 80,253 bp (*R. nigrum*) to 80,439 bp (*R. rubrum*). The overall GC content of the four cp genomes ranged from 38.21% (*R. uva-crispa*) to 38.08% (*R. rubrum*). Moreover, the GC content of LSC, SSC, and IRs ranged from 36.14% (*R. rubrum*) to 36.30% (*R. uva-crispa*), 33.15% (*R. nigrum*) to 33.41% (*R. uva-crispa*), and 43.08% (*R. rubrum*) to 43.13% (*R. uva-crispa*), respectively. Notably, red currant and white currant exhibited an identical cp structure, including total length, length of the quadruplicate parts, and the GC content. A total of 131 genes were commonly annotated in the four cp genomes, including 86 protein-coding genes, 37 tRNA genes, and eight rRNA genes (Table 2). Among them, seven protein-coding genes, seven tRNA genes, and four rRNA genes were duplicated in the IRs. A total of 19 genes (*petB*, *petD*, *atpF*, *ndhA*, two *ndhB*, two *rpl2*, two *rps12*, *rpl16*, *rpoC1*, *rps16*, two *trnA-UGC*, *trnG-GCC*, two *trnI-GAU*, *trnK-UUU*, *trnL-UAA*, and *trnV-UAC*) harbored one intron (Table S1); two genes (*ycf3* and *clpP*) harbored two introns; and *rps12* underwent trans-splicing (Fig. S1).

### Identification of repeat elements among the four *Ribes* cp genomes

Aiming to provide useful information for marker development, the repeat elements in the four cp genomes were annotated. A total of 219 long repeats were identified (Fig. 3A), including tandem, forward, and palindromic repeats but not reverse and complement repeats. Tandem repeats accounted for the largest proportion (44.7%), followed by palindromic repeats (30.1%) and then forward repeats (25.1%) (Tables S2 and S3). For blackcurrant (*R. nigrum*), red/white currant (*R. rubrum*), and gooseberry (*R. uva-crispa*), 53, 51, and 64 long repeats were identified, respectively. A total of 176 SSR loci were identified in the four cp genomes, including mononucleotides (97.2%) and dinucleotides (2.8%). In addition, 54, 40, and 42 SSRs were identified in blackcurrant (*R. nigrum*), red/white currant (*R. rubrum*), and gooseberry (*R. uva-crispa*), respectively (Fig. 3B and Table S4). SSRs showed the highest distribution frequency in the LSC (32–40), followed by the SSC (4–7), and then the IRs (2–4) (Fig. 3C). Moreover, SSRs had the highest distribution frequency in the intergenic region (28–33), followed by the intron region (6–9), and the CDS (3–17) (Fig. 3D).

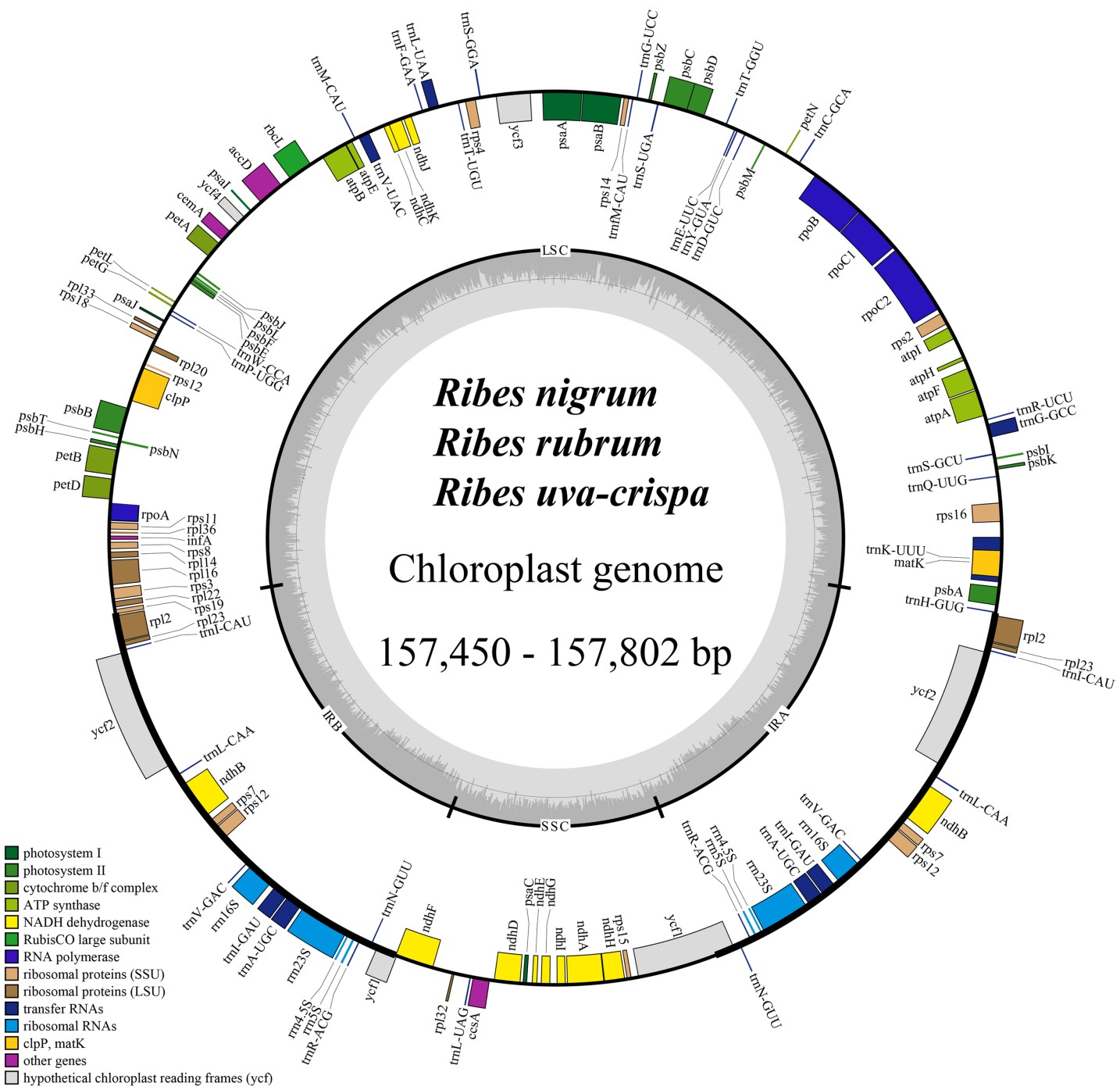

**Figure 2 Consensus feature map of cp genomes for blackcurrant (*Ribes nigrum*), red and white currant (*R. rubrum*), and gooseberry (*R. uva-crispa*).** Genes outside and inside the circle are transcribed clockwise and counterclockwise, respectively. Gray bars in the inner ring show the GC content percentage. The inverted repeat regions (IRa and IRb) are denoted with thick lines, which are separated by the large single copy (LSC) region and the small single copy (SSC) region. Genes belonging to different functional groups are color-coded accordingly at the bottom left.

**Table 1 Summary of cp genome subunits in the four *Ribes* taxa.**

| Species | Blackcurrant (*R. nigrum*) | Red currant (*R. rubrum*) | White currant (*R. rubrum*) | Gooseberry (*R. uva-crispa*) |
|---|---|---|---|---|
| Total size | 157,450 bp | 157,802 bp | 157,802 bp | 157,659 bp |
| LSC length | 87,086 bp | 87,369 bp | 87,369 bp | 87,369 bp |
| SSC length | 18,350 bp | 18,423 bp | 18,423 bp | 18,288 bp |
| IR length | 26,007 bp | 26,005 bp | 26,005 bp | 26,001 bp |
| CDS length | 80,253 bp | 80,439 bp | 80,439 bp | 80,340 bp |
| LSC GC content | 36.19% | 36.14% | 36.14% | 36.30% |
| SSC GC content | 33.15% | 33.19% | 33.19% | 33.41% |
| IR GC content | 43.12% | 43.08% | 43.08% | 43.13% |
| Total GC content | 38.13% | 38.08% | 38.08% | 38.21% |

**Table 2 Gene annotation of the four *Ribes* cp genomes.**

| Function | Gene names | Number |
|---|---|---|
| Photosystem I | *psaA*; *psaB*; *psaC*; *psaI*; *psaJ* | 5 |
| Photosystem II | *psbA*; *psbB*; *psbC*; *psbD*; *psbE*; *psbF*; *psbH*; *psbI*; *psbJ*; *psbK*; *psbL*; *psbM*; *psbN*; *psbT*; *psbZ* | 15 |
| Cytochrome b/f complex | *petA*; *petB**; *petD**; *petG*; *petL*; *petN* | 6 |
| ATP synthase | *atpA*; *atpB*; *atpE*; *atpF**; *atpH*; *atpI* | 6 |
| NADH-dehydrogenase | *ndhA**; *ndhB** (×2); *ndhC*; *ndhD*; *ndhE*; *ndhF*; *ndhG*; *ndhH*; *ndhI*; *ndhJ*; *ndhK* | 12 |
| Large subunit of ribosome | *rpl2** (×2); *rpl14*; *rpl16**; *rpl20*; *rpl22*; *rpl23* (×2); *rpl32*; *rpl33*; *rpl36* | 11 |
| DNA-dependent RNA polymerase | *rpoA*; *rpoB*; *rpoC1**; *rpoC2* | 4 |
| Small subunit of ribosome | *rps2*; *rps3*; *rps4*; *rps7* (×2); *rps8*; *rps11*; *rps12** $^T$ (×2); *rps14*; *rps15*; *rps16**; *rps18*; *rps19* | 14 |
| Transfer RNAs | *trnA-UGC** (×2); *trnC-GCA*; *trnD-GUC*; *trnE-UUC*; *trnF-GAA*; *trnfM-CAU*; *trnG-GCC**; *trnG-UCC*; *trnH-GUG*; *trnI-CAU* (×2); *trnI-GAU** (×2); *trnK-UUU**; *trnL-CAA* (×2); *trnL-UAA**; *trnL-UAG*; *trnM-CAU*; *trnN-GUU* (×2); *trnP-UGG*; *trnQ-UUG*; *trnR-ACG* (×2); *trnR-UCU*; *trnS-GCU*; *trnS-GGA*; *trnS-UGA*; *trnT-GGU*; *trnT-UGU*; *trnV-GAC* (×2); *trnV-UAC**; *trnW-CCA*; *trnY-GUA* | 37 |
| Ribosomal RNAs | *rrn4.5S* (×2); *rrn5S* (×2); *rrn16S* (×2); *rrn23S* (×2) | 8 |
| Conserved open reading frames | *ycf1* (×2); *ycf2* (×2); *ycf3***; *ycf4* | 6 |
| Subunit of acetyl-CoA-carboxylase | *accD* | 1 |
| C-type cytochrome synthesis gene | *ccsA* | 1 |
| Envelop membrane protein | *cemA* | 1 |
| Protease | *clpP*** | 1 |
| Translational initiation factor | *infA* | 1 |
| Maturase | *matK* | 1 |
| Rubisco | *rbcL* | 1 |

**Note:**
* indicates one intron; ** indicates two introns; (×2) indicates that the gene contains two copies in IRs; $^T$ indicates trans-spliced genes.

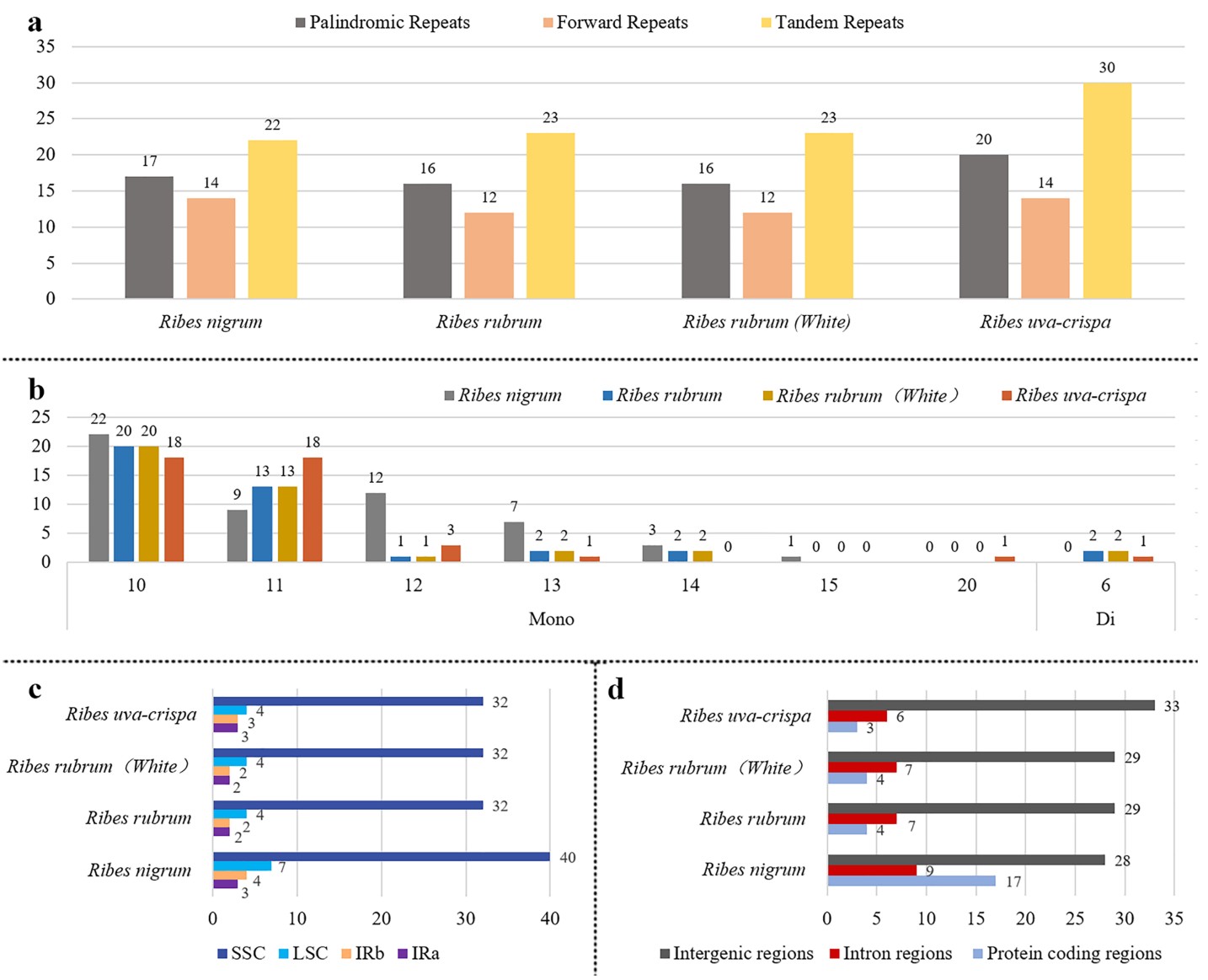

**Figure 3 Characterization of repeats in the four *Ribes* cp genomes.** (A) Summary of long repeats in the four *Ribes* cp genomes. (B) Types and repetitions of SSRs identified in the four cp genomes. Mono: mononucleotide-repeat; Di: dinucleotide-repeat. The numbers above the bars are the counts of the SSRs and those below the bars are the repetitions. (C) Counts of SSRs in the LSC, SSC, and IR regions. (D) Counts of SSRs in intergenic, intron, and protein-coding regions.

## Genomic comparison of the nine *Ribes* cp genomes

We performed a genomic comparison of nine *Ribes* cp genomes, including the four assembled in the present study and five cp genomes released previously (*R. glaciale*, *R. fasciculatum* var. *chinense*, *R. nevadense*, *R. roezlii*, and *R. odoratum*). The nine *Ribes* cp genomes showed identical genomic features, gene orders, and gene numbers (Fig. 4). As expected, the coding regions exhibited lower levels of divergence than non-coding regions. The pairwise genetic divergence between any two of the nine cp genomes was lower than 2.90% (Table S5). The highest sequence divergence was detected between *R. fasciculatum* var. *chinense* and *R. odoratum*. Notably, the cp genomes of red currant and

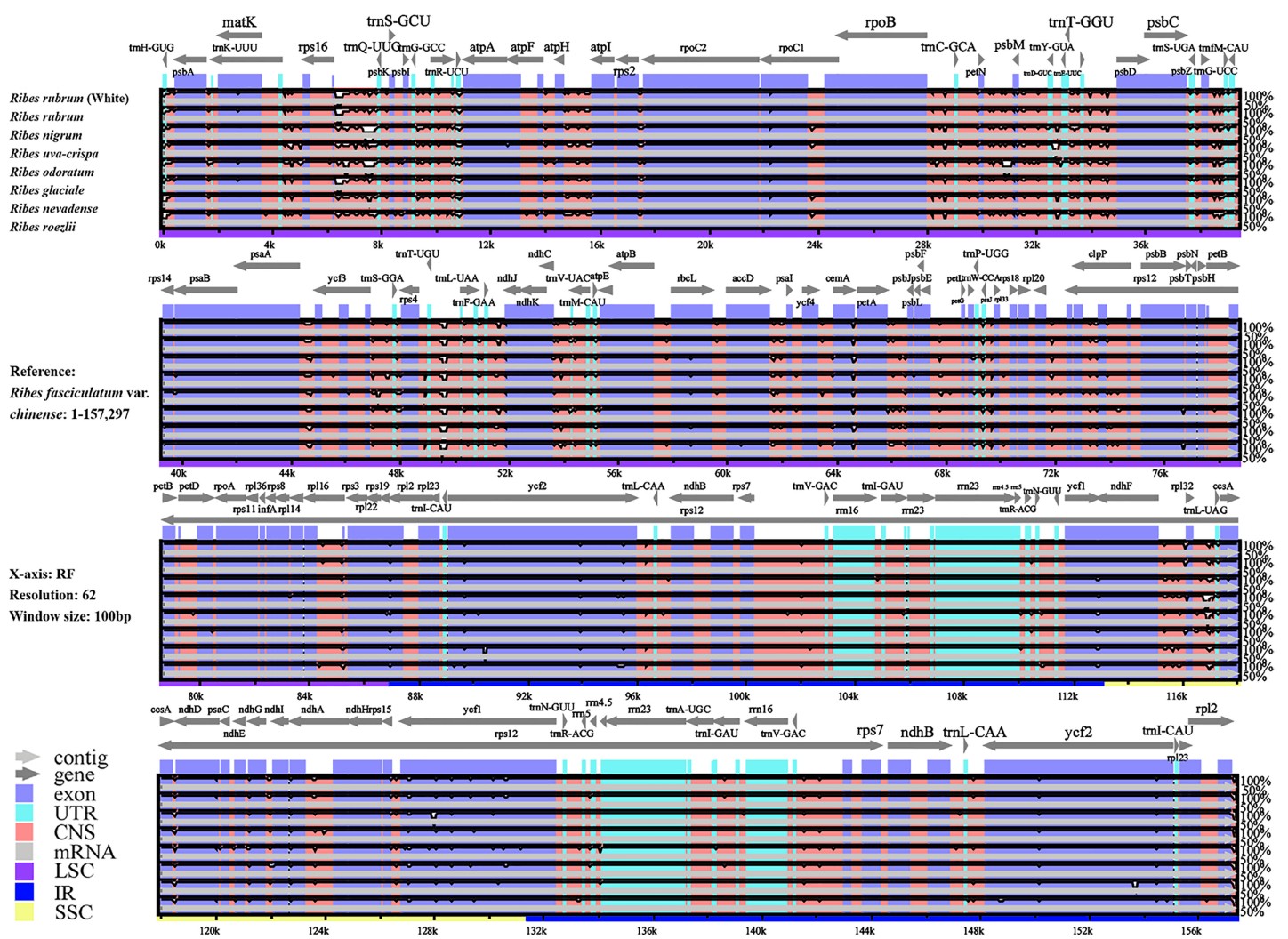

**Figure 4 Global alignment of the nine *Ribes* cp genomes.** The x-axis represents aligned base sequences, and the y-axis represents percent pairwise identity within 50–100%. (Coding regions are shown in blue, and non-coding regions are indicated by pink. The location and direction of each gene are indicated by gray arrows above the alignment. CNS, conserved noncoding sequences; UTR, untranslated region).

white currant showed 100% sequence identity (Dataset S1). The IR/SC boundary shifts among the nine cp genomes were investigated (Fig. 5). Generally, 16 types of IR/SC shift were identified among the nine *Ribes* cp genomes. The *rps*19 gene spanned the LSC-IRb boundary (JLB) with five types of shifts, including "219 bp + 60 bp" (*R. uva-crispa*, *R. nigrum*, *R. roezlii*, and *R. nevadense*), "212 bp + 70 bp" (*R. rubrum*), "209 bp + 70 bp" (*R. glaciale*), "207 bp + 72 bp" (*R. odoratum*), and "202 bp + 77 bp" (*R. fasciculatum* var. *chinense*). The pseudogene gene *ycf1* (Ψ) spanned the IRb-SSC boundary (JSB) with four types of shifts, including "1,131 bp + 11 bp" (*R. uva-crispa* and *R. roezlii*), "1,141 bp + 16 bp" (*R. nigrum*, *R. nevadense*, and *R. glaciale*), "1,131 bp + 98 bp" (*R. rubrum* and *R. fasciculatum* var. *chinense*), and "1,141 bp + 94 bp" (*R. odoratum*). The *ycf1* gene spanned the SSC-IRa boundary (JSA) with four types of shifts, including "4,530 bp + 1,131 bp" (*R. uva-crispa* and *R. roezlii*), "4,406 bp + 1,141bp" (*R. nigrum*), "4,257 bp +

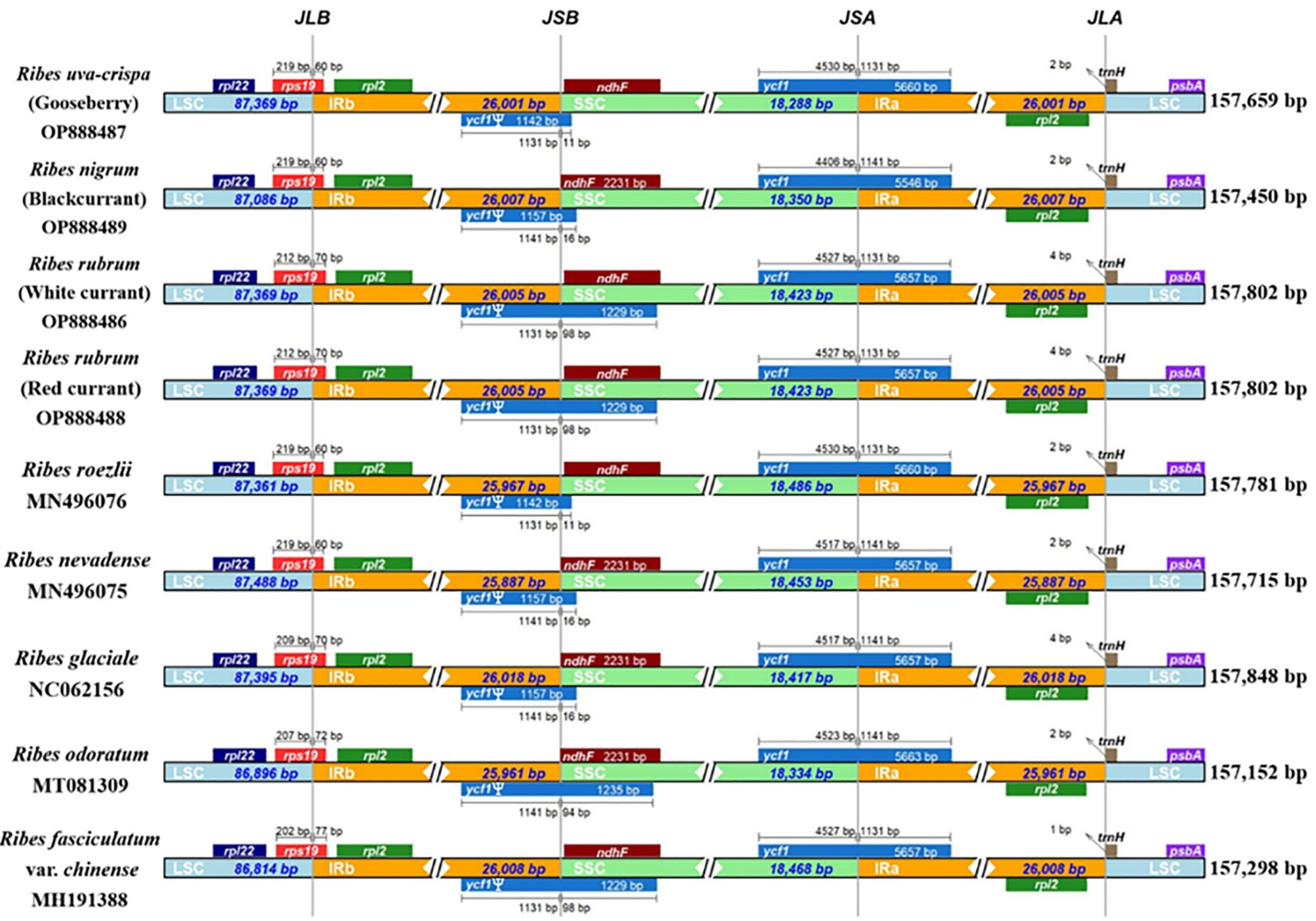

**Figure 5 Comparison of LSC, SSC, and IR region boundaries in the nine *Ribes* cp genomes.** Distance in the figure is not to scale. The gene names are indicated in the colored box, and the length of the corresponding region is shown next to the box. Ψ represents the pseudogene. JLB, LSC-IRb boundary; JSB, IRb-SSC; JSA, SSC/IRa; JLA, IRa-LSC. LSC, large single copy; IRa, IRb, two IR regions that are identical but in opposite orientations; SSC, small single copy.

1,131 bp" (*R. rubrum* and *R. fasciculatum* var. *chinense*), "4,517 bp + 1,141 bp" (*R. nevadense* and *R. glaciale*), and "4,523 bp + 1,141 bp" (*R. odoratum*). The *trnH* gene spanned the IRa-LSC boundary (JLA) with three types of IRa residue, including "1 bp" (*R. fasciculatum* var. *chinense*), "2 bp" (*R. uva-crispa*, *R. nigrum*, *R. roezlii*, *R. nevadense*, and *R. odoratum*), and "4 bp" (*R. rubrum* and *R. glaciale*).

## Identification of SNPs and Indels in the *Ribes* cp genomes

A total of 3,322 SNPs and 485 Indels were identified in the nine *Ribes* cp genomes (Table 3). The LSC region comprised the largest number of SNPs (2,337) and Indels (361), followed by the SSC region (701 SNPs and 65 Indels). Generally, the CDS is more conserved than non-coding regions. Here, a total of 1,594 SNPs and 377 Indels were detected in the intergenic spacer (IGS); 424 SNPs and 80 Indels were found in the intron; and 1,304 SNPs and 28 Indels were detected in the CDS (Table S6). The most polymorphic
**Table 3 Summary of SNPs and Indels identified in the nine *Ribes* cp genomes.**

| Genomic region | SNPs | Indels |
|---|---|---|
| IRa | 141 | 36 |
| IRb | 142 | 35 |
| LSC | 2,337 | 348 |
| SSC | 701 | 65 |
| LSC-IRb | 1 | 0 |
| IRb-SSC | 0 | 1 |
| Total | 3,322 | 485 |
| Intergenic regions | 1,594 | 377 |
| Intron | 424 | 80 |
| CDS | 1,304 | 28 |

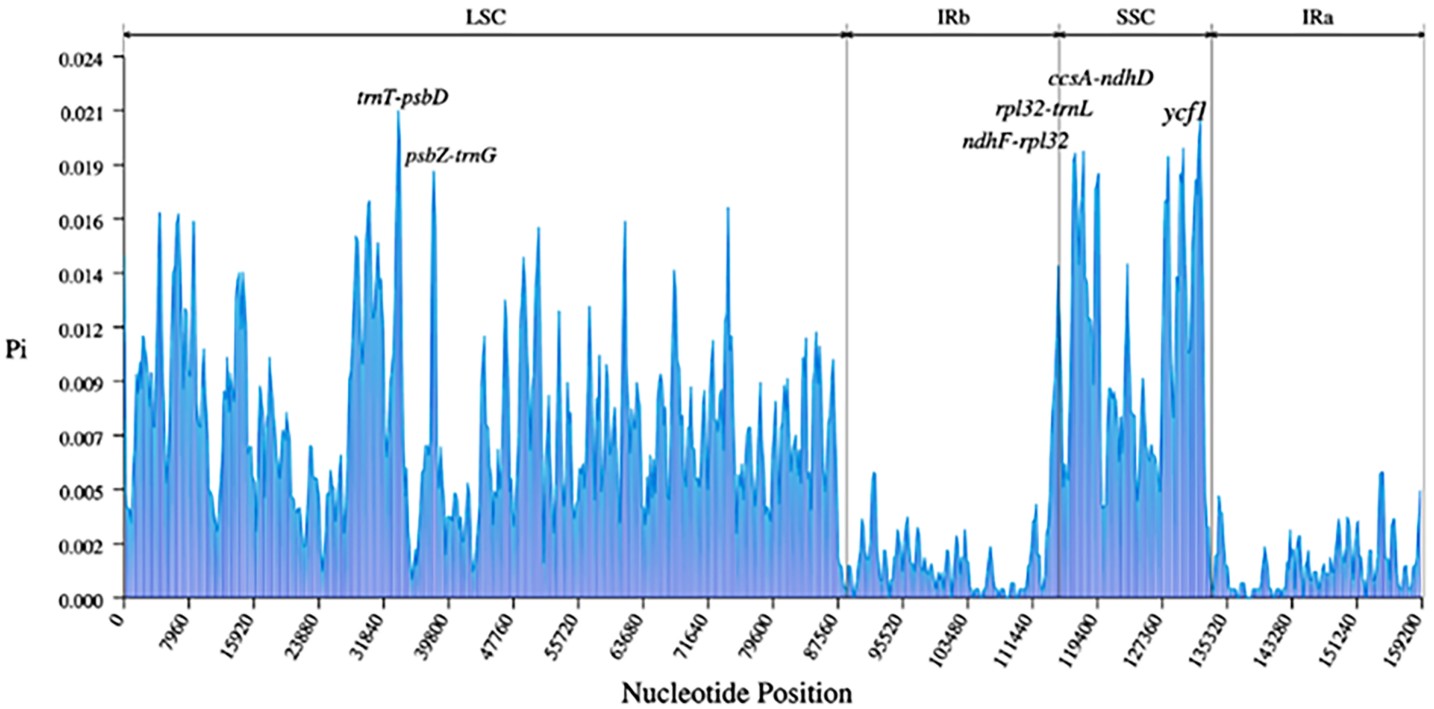

**Figure 6 Nucleotide diversities (*Pi*) of the nine *Ribes* chloroplast genomes presented in a sliding window approach.** X-axis represents the position of the window midpoint; Y-axis represents the nucleotide diversity within each window (window size: 600 bp; step size: 200 bp). Six regions with relatively high *Pi* values were marked based on the alignment and annotation.

intergenic region was *trnT-psbD*, which contains 86 mutation sites, followed by *ndhF-rpl32* (76) and *atpH-atpI* (71). The most variable gene *ycf1*, whose coding-sequence harbors 248 mutation sites, followed by *rpoC2* (104), *ndhF* (56), and *ycf2* (56). In terms of introns, the introns of *trnK* and *matK* contained the most variable mutation sites (57), followed by introns of *rpl16* (38) and *ndhA* (57) (Tables S7 and S8). In the pairwise comparison, the most variation sites were found between *R. roezlii* and *R. fasciculatum* var. *chinense*
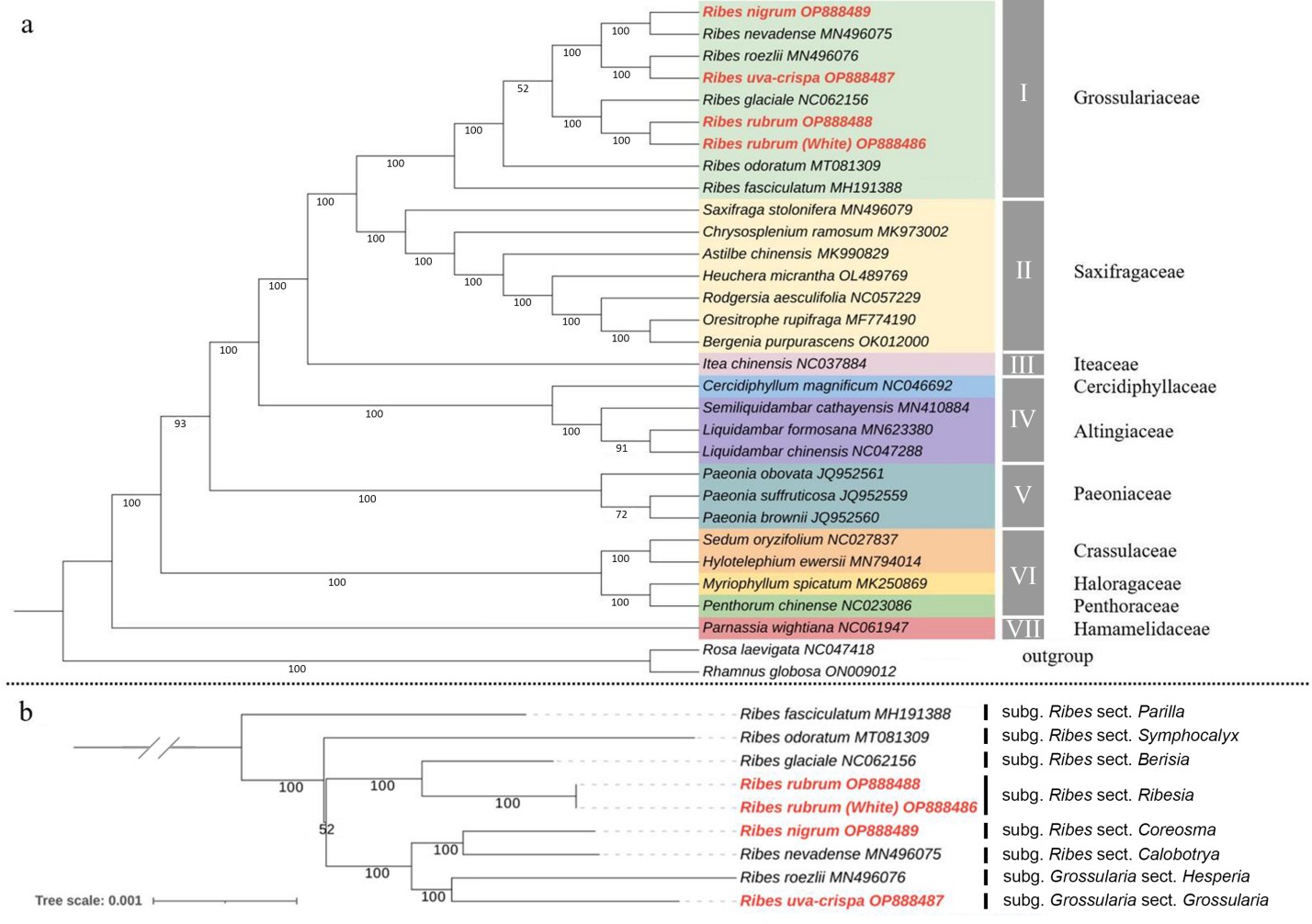

**Figure 7 Chloroplast phylogeny of Saxifragales and *Ribes*.** (A) Maximum likelihood tree constructed using 29 complete cp genomes from Saxifragales and two outgroup species from Rosales with 1,000 bootstrap replications. (B) Maximum likelihood tree inferred from nine complete cp genomes of *Ribes* with 1,000 bootstrap replications. Vertical bars at the right label the section classification of these nine taxa according to the consensus revision of *Berger (1924)*, *Sinnott (1985)*, *Messinger, Hummer & Liston (1999)*, and *Senters & Soltis (2003)*.

(1,863), and the fewest variation sites (507) were detected between blackcurrant (*R. nigrum*) and *R. nevadense* (Table S9). The nucleotide diversity (*Pi*) of the nine *Ribes* cp genomes was visualized using a window length of 600 bp and a step size of 200 bp (Fig. 6). The average *Pi* of whole sequences, SSC, LSC, and IR regions was 0.0060, 0.011, 0.0077, and 0.0015 respectively. Five IGS (*trnT-psbD*, *psbZ-trnG*, *ndhF-rpl32*, *rpl32-trnL*, and *ccsA-ndhD*), and one gene (*ycf1*) exhibited relatively high genetic diversities (*Pi* ≥ 0.018). *TrnT-psbD* was the most polymorphic region with a *Pi* of 0.021.

## Phylogenomic analysis of Grossulariaceae

To determine the phylogenetic position of Grossulariaceae in Saxifragales, ML and Bayesian phylogenetic trees were constructed based on 31 complete cp genomes, including the above-mentioned nine *Ribes*, 20 representative species from Saxifragales and two

outgroup species from Rosales (Table S9). The ML and Bayesian tree topology consensually revealed seven well-supported main clades in Saxifragales (BS ≥ 93% and PP = 100%), including clade I, Grossulariaceae; clade II, Saxifragaceae; clade III, Iteaceae; clade IV, Cercidiphyllaceae and Altingiaceae; clade V, Paeoniaceae; clade VI, Crassulaceae, Haloragaceae, and Penthoraceae; and clade VII, Hamamelidaceae (Figs. 7A and S2). Notably, the nine *Ribes* taxa were clustered independently to form clade I with maximum BS and PP values instead of nesting within Saxifragaceae (clade II), indicating the monophyly of Grossulariaceae. In terms of the *Ribes* genus (Fig. 7B), *R. fasciculatum* (subgenus *Ribes*, section *Parilla*) was first differentiated from other species, followed by *R. odoratum* (subgenus *Ribes*, section *Symphocalyx*). However, the ancestral clade of the remaining seven species was not well-resolved (BS = 52%). In addition, paraphyletic relationships were found between the *Berisia* and *Ribesia* sections, between the *Coreosma* and *Calobotrya* sections, and between the *Hesperia* and *Grossularia* sections. Notably, the *Grossularia* subgenus was nested within the *Ribes* subgenus instead of being parallel to it at the basal of the tree as expected.

## DISCUSSION

The present study for the first time assembled and characterized the complete cp genomes of four cultivated *Ribes* taxa, including blackcurrant (*R. nigrum*), red currant and white currant (*R. rubrum*), and gooseberry (*R. uva-crispa*), which enriches the genomic foundation for *Ribes*. The Grossulariaceae showed relatively high chloroplast conservativeness in terms of sequence length, structural organization, gene content, order, and arrangement than other plant families such as Gentianaceae, Asteraceae, Caprifoliaceae, and the genetic close family, Saxifragaceae (*Walker, Zanis & Emery, 2014*; *Sun et al., 2018*; *Wang et al., 2020*; *Chen et al., 2022*). Intriguingly, the results showed that the IR/SC shift types of the nine *Ribes* cp genomes are in accordance with the phylogenetic groups (Figs. 5, 7B and S2), suggesting the shrinkage and expansion of the IR/SC boundaries might serve as an additional confirmation for phylogenetic results that calculated from the algorithms based on sequences. Notably, the cp genomes of red currant (OP888488) and white currant (OP888486) are 100% identical, which supports the hypothesis: white currant is an albino cultivar of red currant (*Reisch & Pratt, 1996*; *Prange, 2002*; *Lim, 2012*) and calls for further verification based on nuclear data. Repeat elements play an important role in sequence divergence and are involved in plastome rearrangement (*Timme et al., 2007*; *Weng et al., 2014*). Herein, 51/40, 64/42, and 53/54 long-repeats/SSR-loci was respectively identified in *R. rubrum*, *R. uva-crispa*, and *R. nigrum* cp genomes, whereas that of the wild *R. odoratum* is 49/56 (*Wang et al., 2021*). These differences not merely display the characteristic evolutionary trace of these plastomes but also provide effective information for species discrimination and population genetics. Moreover, a total of 3,322 SNPs and 485 Indels were identified in the nine released *Ribes* cp genomes. In particular, the intergeneric spacer of *trnT-psbD*, the CDS of *ycf1*, and the introns of *trnK* and *matK* showed high-resolution potentials. The molecular markers developed from these loci may contribute to genetic evaluations for germplasm collections, conservation genetic research for endangered species or enigmatic taxa, DNA barcoding for endemic

species or important cultivars, and hybrid identification in cross-breeding of *Ribes* in the future.

In terms of the phylogeny of Grossulariaceae, we for the first time used the genomic data of *Ribes* species to clarify the monophyly of Grossulariaceae and the paraphyletic relationship between Grossulariaceae and Saxifragaceae, which are consistent with the results of previous studies about the phylogeny of Saxifragales (*Dong et al., 2013*, *2018*). Moreover, in both ML and Bayesian trees, the highly supported co-ancestral clade of Saxifragaceae and Grossulariaceae and the relatively later divergence between them were revealed, suggesting their close phylogenetic relationship, which is in accordance with the previous studies based on various data types (*Dong et al., 2013*; *Soltis et al., 2013*; *Han et al., 2022*). In terms of the infrageneric relationships, our results generally support the consensus classification of *Ribes* proposed by *Berger (1924)* and revised by *Sinnott (1985)*, that is, the *Ribes* genus includes 12 sections of *Berisia*, *Calobotrya*, *Coreosma*, *Grossularioides*, *Heritiera*, *Parilla*, *Ribesia*, *Symphocalyx*, *Grossularia*, *Hesperia*, *Lobbia* and *Robsonia*, which is also supported by recent studies based on molecular markers (*Messinger, Hummer & Liston, 1999*; *Weigend, Mohr & Motley, 2002*; *Senters & Soltis, 2003*; *Schultheis & Donoghue, 2004*). Although our results revealed a well-supported clade of the subgenus *Grossularia*, only two species (*R. roezlii* and *R. uva-crispa*) in this subgenus were used. Therefore, the monophyletic origin of *Grossularia* cannot be asserted here. However, in any case, the clade of the *Grossularia* subgenus is nested within the *Ribes* subgenus and shows a paraphyletic relationship with the co-ancestor of the *Calobotrya* and *Coreosma* sections either in the present study or in previous studies (*Senters & Soltis, 2003*; *Schultheis & Donoghue, 2004*), which is incongruent to that of the generally accepted subclassification based on morphological characteristics of the *Ribes* genus, regardless of whether gooseberry (*Grossularia*) is monophyletic. However, according to the limited samples and only cp genomic data used in the present study, the thorough clarification of intrageneric phylogeny and infrageneric resolution, evolutional scenario, and diversification history of *Ribes* still call for comprehensive studies based on large-scale sample collection, nuclear genomic data, biogeographic information, and fossil evidence.

## CONCLUSIONS

The complete cp genomes of blackcurrant (*R. nigrum*), red currant and white currant (*R. rubrum*), and gooseberry (*R. uva-crispa*) have lengths of 157,450–157,802 bp and share a total of 131 genes (86 protein-coding genes, 37 tRNA genes, and eight rRNA genes). Notably, a 100% sequence identity was observed between red currant and white currant, supporting the inference that the latter is an albino cultivar of the former. A total of 485 Indels and 3,322 SNPs were identified in the nine released *Ribes* cp genomes. *Ycf1* ($Pi = 0.020$) and *trnT-psbD* ($Pi = 0.021$) are the most polymorphic genic and intergenic regions, respectively. The ML and Bayesian phylogenetic trees provide strong support for the monophyly of Grossulariaceae in Saxifragales and the paraphyletic relationship between Saxifragaceae and Grossulariaceae. The tree topology of the nine *Ribes* taxa shows that the *Grossularia* subgenus is nested within the *Ribes* subgenus, and has a paraphyletic

relationship with the co-ancestor of the *Calobotrya* and *Coreosma* sections. Our findings lay a foundation for the phylogenetic research and molecular breeding of *Ribes* species.

## ACKNOWLEDGEMENTS

We are grateful to Professor Yonghe Zhang of Northeast Agricultural University for his introduction about the plant materials used in this study.

### Funding
This work was funded by the China Postdoctoral Science Foundation (Grant Number 2021MD703804), the National Natural Science Foundation of China, (Grant Number 32102332), the Natural Science Foundation of Heilongjiang Province, (Grant Number LH2021C031) and the National Key R&D Program of China, (Grant Number 2022YFD1600500). The funders had no role in study design, data collection and analysis, decision to publish, or preparation of the manuscript.

### Grant Disclosures
The following grant information was disclosed by the authors:
China Postdoctoral Science Foundation: 2021MD703804.
National Natural Science Foundation of China: 32102332.
Natural Science Foundation of Heilongjiang Province: LH2021C031.
National Key R&D Program of China: 2022YFD1600500.

### Competing Interests
The authors declare that they have no competing interests.

### Author Contributions
- Xinyu Sun analyzed the data, prepared figures and/or tables, and approved the final draft.
- Ying Zhan performed the experiments, prepared figures and/or tables, and approved the final draft.
- Songlin Li analyzed the data, prepared figures and/or tables, and approved the final draft.
- Yu Liu performed the experiments, prepared figures and/or tables, and approved the final draft.
- Qiang Fu performed the experiments, prepared figures and/or tables, and approved the final draft.
- Xin Quan performed the experiments, prepared figures and/or tables, and approved the final draft.
- Jinyu Xiong analyzed the data, prepared figures and/or tables, and approved the final draft.
- Huixin Gang performed the experiments, prepared figures and/or tables, and approved the final draft.

- Lijun Zhang performed the experiments, prepared figures and/or tables, and approved the final draft.
- Huijuan Qi conceived and designed the experiments, prepared figures and/or tables, and approved the final draft.
- Aoxue Wang conceived and designed the experiments, authored or reviewed drafts of the article, and approved the final draft.
- Junwei Huo conceived and designed the experiments, authored or reviewed drafts of the article, and approved the final draft.
- Dong Qin conceived and designed the experiments, authored or reviewed drafts of the article, funding, and approved the final draft.
- Chenqiao Zhu conceived and designed the experiments, analyzed the data, authored or reviewed drafts of the article, funding, and approved the final draft.

### DNA Deposition

The following information was supplied regarding the deposition of DNA sequences:

The four complete chloroplast assembly sequences are available at GenBank: OP888489 (blackcurrant), OP888488 (red currant), OP888486 (white currant), and OP888487 (gooseberry).

### Data Availability

The sequencing data is available at NCBI SRA: SRR23008392 (blackcurrant, *R. nigrum*), SRR22955278 (red currant, *R. rubrum*), SRR23008393 (white currant, *R. rubrum*) and SRR22956007 (gooseberry, *R. uva-crispa*).

### Supplemental Information

Supplemental information for this article can be found online at http://dx.doi.org/10.7717/peerj.16272#supplemental-information.

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
