# Peer review of "Complete chloroplast genome assembly and phylogenetic analysis of blackcurrant (Ribes nigrum), red and white currant (Ribes rubrum), and gooseberry (Ribes uva-crispa) provide new insights into the phylogeny of Grossulariaceae"

_PeerJ, doi:10.7717/peerj.16272_

## Round 0.1 · original submission · Major Revisions

The reviewers have now completed the assessment of your manuscript. Detailed review reports have been generated. Please go through the comments and revise your manuscript based on the reports accordingly. A point-by-point response letter is required during the submission of your revised manuscript for reconsideration by the journal.

Reviewer 1 ·

Basic reporting

The manuscript describes on the characterization, comparative analysis and phylogenomic reconstruction of the complete chloroplast genomes of four selected Ribes taxa. I am not in a position to perform English check, but obviously, the manuscript is well-written in terms of grammar. The contents are tidy and well-organized; however, the authors tend to put discussion materials in the Result section.The main drawback of this work is the phylogenetic analysis (refer comments below). The objectives and findings are relevant and this work should be accepted after corrections are made.

Experimental design

The experimental design is sufficient to address the objective of the study.

Validity of the findings

The authors need to be informed with several things:
Abstract
In the results section, red and white currant have 100% identical cp genomes suggesting that white currant should be henceforward designated as R. rubrum var. alba to avoid ambiguity. This claim has no scientific backups. Note that to claim for a variation, color is not a solid character. This can only be regarded as a type of color variant, not a variation.
Also, by referring to just a few of selected species from some selected genera, as well as a limited approach (only one type of dataset, and one tree method used) for phylogenetic analysis, it is not convincing to tell that the subclassification requires revision. The authors should just state that the findings are incongruent to that of the generally accepted subclassifcation via morphological characteristics.

L178 four taxa assembled in present study. There are only 4 taxa used in this study, derived from 3 species.
L182 Although it might not be applicable to all researchers, for me, using MEGA7 to construct an ML tree for cp genome datasets is less efficient. The authors should at least opt for the use of RAxML or IQ-tree, coupled with Bayesian inference using MrBayes to reveal a better molecular placement of the species involved. Furthermore, with the low resolution in certain branch nodes demonstrated in the phylogenetic tree, I would suggest the use of CDS dataset to reconstruct the phylogenetic tree too.
L186 four Ribes taxa
L206-208 This is discussion material
L224-225 This is discussion material
L230-231 This is discussion material
L268 This is discussion material
L271 please recalculate the number of species involved, considering that it could be number of taxa you are referring to at this point
L274 for MEGA, the bootstrap support should be in percentage. please check.
L279-282 The judgement to call a sister lineage in a phylogenetic tree is not possible when only one method (in this case, ML) is used. For molecular placement to be visible, involving at least two different methods is a minimum requirement to justify their positions in the phylogenetic tree.
L283-286 this is discussion material. Also, please read the comments I gave before this on asking for a revision to a generally accepted classification.
L289 this is discussion material.
L295 this as well
L330 proposal is not valid at this point of findings obtained in this study
L337 This suggestion is too extreme at this point due to the findings provided are not convincing.
L342 The use of R. rubrum var. alba is invalid at this point. avoid using this name. This further reflects the suggestions provided in L347-348.

Additional comments

A supporting genetic evidence that could suggest a revision is most likely based on both the plastid and nuclear data, complementing to morphological characteristics. In this case, despite chloroplast data is provided, nothing was reported on the nuclear part and morphological characteristics, it is not wise to suggest or propose a revision. The authors should be informed on the usual practice for species classification prior to presenting such claims in this work. Therefore, this work only characterizes and report on the phylogenetic tree of the 4 selected Ribes taxa based on the complete chloroplast genome sequence.

Reviewer 2 ·

Basic reporting

There are minor changes/information that can be included.
I have the following comments to improve the manuscript's writing:
1. The author can go into further detail about the economics of Ribes species.
2. The significance of chloroplasts in plant growth is emphasized. It should be noted that plastids and chloroplasts are essential for plants. They are plant organelles that, in addition to photosynthesis, perform a variety of other tasks in plant metabolism. It is unclear why plastid genomes exist in the absence of this information. Furthermore, greater information about the appropriateness of plastid genome sequences for species discrimination would be beneficial.
3. I believe the author should consider condensing or deleting some non-essential information to keep the introduction concise.

Experimental design

1. Additional methods, like as MP and Bayes, should, in my opinion, be included in phylogenetic analysis.

Validity of the findings

1. Only the ML tree was recovered. Other approaches, like as MP and Bayes, should be incorporated in phylogenetic analysis, in my opinion.
2. On figure 3, simply change the information in the middle of the picture about sequence length to "range to range." example ribes 157,450bp - 157,802bp"
3. The author can add to a discussion of how genomic analysis of repeat elements (tandem repeats, palindromic repeats, and forward repeats) contributes to our understanding of Ribes evolution and genetic diversity.
4. The authors can consider to include a brief explanation of the research significance for Ribes species improvement, genetic conservation, and breeding.

Additional comments

The paper "Complete chloroplast genome assembly and phylogenetic analysis of blackcurrant (Ribes nigrum), red and white currant (Ribes rubrum), and gooseberry (Ribes uva-crispa) provide new insights into the phylogeny of Grossulariaceae" highlights the significance of genetic information for DNA barcoding applications. The presentation of the text is clear, and the results are honestly convincing.

---

## Round 0.2 · Minor Revisions

Please follow the reviewer's comments and revise the manuscript carefully and thoroughly. We look forward to receiving your revised manuscript soon. Thanks for the efforts.

Reviewer 1 ·

Basic reporting

Substantial changes have been identified in the revised manuscript. Most of my comments have been attained; however, some careless mistakes are found in the text.
1. Numbers less than 10 should be spelled in full
2. Citations should be placed in ascending orders.
3. Some references are incomplete.

Overall, the manuscript can be accepted after the above-mentioned comments are dealt with.

Experimental design

na

Validity of the findings

na

Additional comments

na

---

## Round 0.3 · Minor Revisions

Please fully address the reviewer's questions and provide your revised manuscript as soon as possible to the editorial office.

Reviewer 1 ·

Basic reporting

The authors fail to attain my second comment on "2. Citations should be placed in ascending orders." Let me make this clear:

L90-91. The citation should be Coville et al. 1908, Berger et al. 1924, Sinnott 1985, Poyarkova 1939, Messinger et al.1999. I believe this is applicable to most journals. Please check through and amend them accordingly.

Experimental design

na

Validity of the findings

na

Additional comments

na

---

## Round 0.4 · accepted · Accept

The authors have solved the reviewer's question. Based on reviewers' comments and the authors' revisions, the manuscript is acceptable for publication now.